# SEMI-3DETR: SEMI-SUPERVISED DETECTION TRANSFORMER FOR 3D OBJECT DETECTION

## ABSTRACT

DETR-based 3D detectors have recently emerged as a popular alternative to voting- and voxel-based methods, which offer end-to-end set prediction without hand-crafted priors or voxelization. However, they remain unexplored under semi-supervision, where the scarcity of annotated 3D data impedes their widespread adoption. In this work, we present Semi-3DETR, the first framework to systematically adapt DETR to semi-supervised 3D object detection by addressing challenges unique to 3D. Compared to 2D semi-DETR, semi-supervised 3D DETR faces amplified issues of fragile volumetric pseudo-labels, unstable query alignment, and noisy bipartite matching. Our Semi-3DETR mitigates these issues by introducing three core components: Robust Pseudo-Label Denoising (RPLD) to filter and refine volumetric pseudo-labels against orientation and depth errors, Query Alignment Consistency (QAC) to stabilize teacher–student query correspondence under 3D transformations, and a Hybrid Matching Strategy (HMS) to balance one-to-one and one-to-many assignments under noisy supervision. We further adopt a softmax classifier to enforce class exclusivity and improve pseudo-label reliability in semantically ambiguous 3D categories. Extensive experiments on ScanNet and SUN RGB-D demonstrate the feasibility of our Semi-3DETR with promising results compared to fully supervised and semi-supervised baselines. The source code will be released upon paper acceptance.

## 1 INTRODUCTION

3D object detection from point clouds is a core problem in computer vision with wide applications in autonomous driving, AR/VR, and robotic navigation. The task requires predicting oriented 3D bounding boxes and semantic labels in complex and cluttered environments. Despite making remarkable progress, supervised 3D detectors rely heavily on large-scale annotated 3D datasets whose construction is prohibitively costly due to the time and labor needed to annotate 3D scenes. Semi-supervised 3D object detection (SS3D) offers a promising alternative by leveraging a small labeled subset together with abundant unlabeled 3D data.

Recent progress in SS3D has largely focused on two families of detectors: 1) ***Voting-based methods*** such as VoteNet Qi et al. (2019), which exploit geometric priors to predict object centroids. These priors enable robust pseudo-labeling and have led to strong semi-supervised extensions Zhao et al. (2020); Wang et al. (2021). However, their reliance on handcrafted designs restricts scalability. 2) ***Voxel-based methods*** Rukhovich et al. (2022); Wang et al. (2022a), which discretize point clouds for efficient CNN processing. Recent work DQS3D Gao et al. (2023) introduces quantization-aware consistency to achieve state-of-the-art semi-supervised performance. However, voxelization imposes high computational cost and resolution constraints. In contrast, ***DETR-based methods*** Misra et al. (2021); Liu et al. (2021); Shen et al. (2023) formulate detection as a set prediction using transformers. This removes the need for hand-crafted priors or voxel grids. Despite these advantages, no prior work has explored semi-supervision for 3D DETR-based detection. This gap motivates our work on Semi-3DETR, the first framework to systematically extend DETR to semi-supervised 3D detection.

We believe that it is both timely and important to explore semi-supervised DETR in 3D object detection. As DETR-based models are increasingly adopted for their end-to-end and prior-free formulation, giving them the ability to leverage unlabeled data is crucial for scaling beyond the

limits of annotation. However, unlike voting or voxel detectors, query-based set prediction in DETR introduces distinctive challenges under semi-supervision that remain unaddressed.

In 2D detection, Semi-DETR Zhang et al. (2023) demonstrated that DETR struggles under semi-supervision due to unstable bipartite matching, query misalignment, and pseudo-label noise. They introduced specialized solutions for 2D detection. However, these solutions do not directly generalize to 3D detection, where the challenges are amplified: 1) *Fragile volumetric pseudo-labels:* Small depth or orientation errors in 3D boxes cause significant IoU drops, which make pseudo-label training less reliable. 2) *3D query grounding instability:* Queries combine positional and content embeddings, which are sensitive to geometric transformations and lack deterministic teacher-student alignment. 3) *Matching under noise:* One-to-one matching in 3D is more fragile due to class ambiguity, and sigmoid-based classification magnifies label uncertainty. To address these challenges, we propose Semi-3DETR, a framework with three key innovations for DETR-based semi-supervised 3D detection: 1) *Robust Pseudo-Label Denoising (RPLD)* to mitigate volumetric fragility by uncertainty-aware filtering and box refinement. 2) *Query Alignment Consistency (QAC)* to stabilize teacher-student correspondence in the 3D query space. 3) *Hybrid Matching Strategy (HMS)* that balances one-to-one and one-to-many assignments to ensure robustness to noisy supervision.

In summary, our work makes the following contributions:

- We identify and analyze the unique challenges of semi-supervised DETR in 3D object detection, contrasting them with both semi-2DETR and semi-supervised 3D voting/voxel methods.
- We propose Semi-3DETR, the first semi-supervised framework for 3D DETR-based detection.
- We introduce novel modules: RPLD, QAC, and HMS that directly address query alignment, pseudo-label fragility, and matching instability in 3D.
- Results across multiple 3D detection benchmarks suggest the feasibility of Semi-3DETR compared to fully supervised and semi-supervised baselines.

## 2 RELATED WORK

**3D Object Detection.** Recent 3D object detection methods on point clouds can be broadly categorized into three paradigms. *Voting-based methods* such as VoteNet Qi et al. (2019) predict object centroids directly from point clouds and aggregate local features to generate proposals. Extensions such as H3DNet Zhang et al. (2020) refine bounding boxes using geometric primitives, and RBGNet Wang et al. (2022b) employs ray-based grouping to encode surface geometry. Although effective, these methods rely heavily on hand-crafted geometric priors, which limit generality and scalability, especially in semi-supervised settings. *Voxel-based methods* Zhou & Tuzel (2018); Gwak et al. (2020); Rukhovich et al. (2022); Wang et al. (2022a) discretize point clouds into voxel grids for 3D CNNs. Although sparse convolutions Gwak et al. (2020) alleviate the memory bottleneck, voxelization often sacrifices geometric fidelity. More recent work such as FCAF3D Rukhovich et al. (2022) adopts a fully anchor-free and voxel-based design. Despite achieving state-of-the-art fully supervised results, they still depend on dense annotations. *DETR-based methods* leverage transformers to avoid explicit point grouping or voxelization and preserve global geometric structure. 3DETR Misra et al. (2021) demonstrates the feasibility of end-to-end detection with minimal 3D-specific priors, GroupFree Liu et al. (2021) enhances query refinement through transformer decoding, and V-DETR Shen et al. (2023) introduces architectural improvements that push performance of DETR-based 3D object detection to a new level.

**Semi-supervised 3D Object Detection.** Several semi-supervised 3D object detection methods have been proposed to leverage abundant unlabeled data and reduce the annotation burden. SESS Zhao et al. (2020) is the first point cloud-based semi-supervised detector, which learns from unlabeled data by enforcing consistency across augmentations. 3DIoUMatch Wang et al. (2021) introduces confidence-based filtering through a 3D IoU estimation module to improve pseudo-label quality. Diffusion-based methods further advance this line: Diffusion-SS3D Ho et al. (2023) formulates detection as a denoising process, and Diff3DETR Deng et al. (2025) integrates a transformer with an agent-based diffusion model for query generation. DQS3D Gao et al. (2023) extends FCAF3D to the semi-supervised regime with quantization-aware consistency and dense voxel matching and achieves strong results on indoor benchmarks. Despite numerous advances in semi-supervised 3D object

detection, no method has explored the DETR-based paradigm. Our Semi-3DETR fills this critical gap towards an semi-supervised end-to-end and prior-free 3D detection that avoids handcrafted voting heuristics and discretized voxel grids.

Note that Diff3DETR is not a true DETR-based semi-supervised method. Although it employs a transformer backbone, it retains the VoteNet-style prediction paradigm and loss functions instead of the core DETR principles such as set-based prediction with Hungarian matching. In contrast, our Semi-3DETR is the first to fully extend the DETR framework to semi-supervised 3D object detection. We preserve the end-to-end set prediction paradigm of DETR and directly address the unique challenges of semi-supervised 3D detection, including volumetric noise amplification and the lack of query correspondence between teacher and student models.

## 3 OUR METHODOLOGY

**Problem Definition.** Given a point cloud set of a 3D scene as input, the goal of 3D object detection is to recognize and localize objects of interest within the point cloud, represented by oriented 3D bounding boxes and semantic class labels, respectively. In the semi-supervised learning setting, $N_l$ labeled point clouds $\mathcal{P}^L = \{\mathbf{x}_i^L, \mathbf{y}_i^L\}_{i=1}^{N_l}$ and $N_u$ unlabeled point clouds $\mathcal{P}^U = \{\mathbf{x}_i^U\}_{i=1}^{N_u}$ are available during training, where $\mathbf{x}_i$ denotes the point cloud of a 3D scene and $\mathbf{y}_i^L$ denotes the ground truth annotations for objects of interest in the labeled point cloud $\mathbf{x}_i^L$.

**Overview.** Figure 1 shows the overall framework of our Semi-3DETR, which is based on the teacher–student paradigm with V-DETR Shen et al. (2023) as baseline. The student is updated by backpropagation while the teacher is the EMA of the student. Labeled data provide supervised signals, and unlabeled data with different augmentations enable consistency training. To address the unique challenges of semi-supervised 3D DETR, Semi-3DETR introduces: 1) *Robust Pseudo-Label Denoising (RPLD)* with a query feature-driven IoU head for reliable volumetric pseudo-labels; 2) *Query Alignment Consistency (QAC)* with a query consistency module for stable teacher–student query correspondence; 3) *Hybrid Matching Strategy (HMS)* with a matching degeneration strategy and softmax classifier for robust assignment under noise. These components jointly enable effective semi-supervised learning for 3D DETR-based detection.

### 3.1 ROBUST PSEUDO-LABEL DENOISING (RPLD)

The absence of full supervision in DETR-based semi-supervised 3D object detection often results in low-quality or duplicated proposals. Unlike 2D Semi-DETR Zhang et al. (2023) where pseudo-labels can be refined using cost-based box assignment, 3D pseudo-labels are far more fragile: small errors in orientation, scale, or depth lead to sharp drops in volumetric IoU. This makes simple confidence-based filtering unreliable in 3D because a bounding box with a high classification score may still be geometrically implausible. To address this unique challenge, we introduce a **query feature-driven 3D IoU estimation head**, which directly leverages the semantic and geometric information encoded in the decoder query features for robust pseudo-label denoising. In parallel with the detection head of V-DETR Shen et al. (2023), our IoU head predicts class-aware 3D IoU for each proposal to provide a *localization confidence* signal that complements classification confidence. This design explicitly grounds pseudo-label filtering in 3D geometry for the suppression of poorly localized and duplicated predictions that would otherwise misguide semi-supervised training.

Our IoU estimation head is a 3-layer perceptron with the ReLU activation function and hidden dimension, and a linear projection layer. For each 3D proposal, the IoU estimation head predicts its 3D IoU with respect to its corresponding ground truth bounding box, and then each 3D IoU prediction is selected by the semantic class label. Let $V = \{v\}$ denotes the 3D IoU ground truths of the 3D proposals, and $\hat{V} = \{\hat{v}\}$ denotes 3D IoU predicted by the IoU estimation head. We apply the 3D IoU between the predicted bounding boxes and their corresponding ground truth bounding boxes as the 3D IoU ground truths $V$. The 3D IoU predictions of the IoU estimation head and the 3D IoU ground truths are compared in the Huber loss function given by:

$$\mathcal{L}_{\text{iou}}(\hat{v}, v) = \begin{cases} \dfrac{1}{2}(\hat{v} - v)^2, & \text{if } |\hat{v} - v| \leq \delta \\ \delta|\hat{v} - v| - \dfrac{1}{2}\delta^2, & \text{if } |\hat{v} - v| > \delta \end{cases}, \tag{1}$$

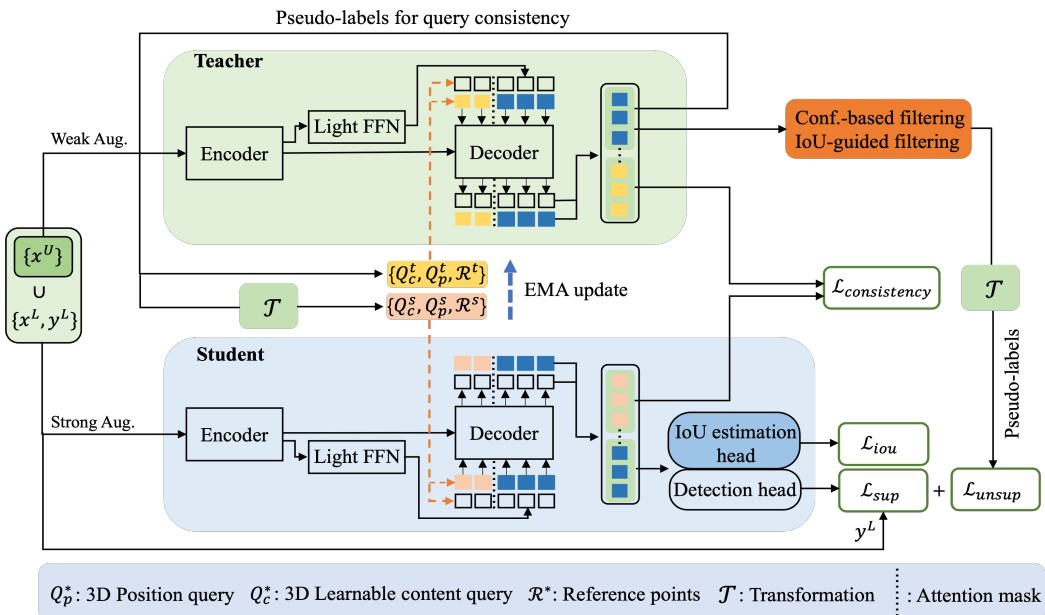

Figure 1: Overview of our Semi-3DETR. Our framework introduces 3D-specific innovations: a ***query feature-driven IoU head*** to filter noisy pseudo-labels, a ***query consistency module*** that reconstructs object queries for semantic and geometric alignment, a ***matching degeneration strategy*** (one-to-many pre-training → one-to-one semi-supervision) to stabilize learning, and a ***softmax classifier*** to ensure reliable pseudo-label selection.

where $\delta = 1.0$. Our Semi-3DETR trains in two stages. In the pre-training stage, we jointly train V-DETR with the IoU estimation head on labeled data (Eq. 7). In the semi-supervised stage, the IoU predictions are exploited to refine pseudo-labels on unlabeled data (Eq. 9). Specifically, a prediction is retained only if it satisfies *both* i) a classification confidence threshold $\tau$ to ensure semantic correctness, and ii) a 3D IoU threshold predicted by our head to ensure precise localization.

By coupling semantic confidence with query-driven geometric confidence, our Semi-3DETR introduces a filtering mechanism uniquely suited to 3D. This departs from the cost-based pseudo-label mining in the 2D Semi-DETR, and establishes a principled way to suppress noisy pseudo-labels in the more error-sensitive 3D domain.

## 3.2 QUERY ALIGNMENT CONSISTENCY (QAC)

The core idea of query alignment consistency is to enforce agreement between predictions from the teacher and student models to encourage learning of structural and semantic invariances. Each object query in a DETR-based 3D detector corresponds to a specific instance and uses cross-attention to extract relevant features from the encoder, which are then used to predict the bounding box and semantic label. From this perspective, object queries should be sufficiently numerous and consistently maintained between teacher and student models to ensure effective and reliable matching. However, unlike CNN-based semi-supervised frameworks where region-level features can be directly aligned, DETR-based frameworks face a critical challenge: the object queries of the teacher and student models do not naturally correspond, and therefore making consistency-based regularization difficult.

To address this issue, we propose a **query consistency-based regularization scheme** that leverages pseudo-labels generated by the teacher model. Our approach constructs a set of specialized 3D object queries that are explicitly shared between teacher and student models to ensure better alignment and consistency. Unlike Semi-DETR Zhang et al. (2023), which enforces cross-view consistency only on 2D image queries, our method reconstructs 3D object queries that capture both semantic features and geometric attributes of center, size, and orientation. This design is critical in 3D where even small spatial perturbations can cause large volumetric discrepancies that cannot be resolved by 2D consistency methods.

Concretely, we decouple the query consistency branch from the original detection branch in the transformer decoder. For each unlabeled sample, pseudo-labels from the teacher are used to construct specialized queries composed of a position query $Q_p$, a content query $Q_c$, and reference points $\mathcal{R}$. The position query $Q_p$ is computed from the pseudo box center $\hat{\mathbf{c}}$ and size $\hat{\mathbf{s}}$ using a multilayer perceptron F with two linear layers. Reference points $\mathcal{R}$ are obtained by combining the pseudo box center $\hat{\mathbf{c}}$, size $\hat{\mathbf{s}}$, and rotation angle $\hat{\mathbf{a}}$. The content queries $Q_c$ are initialized with the object query features $\hat{f}$ predicted by the teacher decoder. Formally:

$$Q_p^t = \mathrm{F}([\hat{\mathbf{c}}, \hat{\mathbf{s}}]), \quad Q_p^s = \mathrm{F}([\mathcal{T}(\hat{\mathbf{c}}), \mathcal{T}(\hat{\mathbf{s}})]), \quad \mathcal{R}^t = [\hat{\mathbf{c}}, \hat{\mathbf{s}}, \hat{\mathbf{a}}],$$
$$\mathcal{R}^s = [\mathcal{T}(\hat{\mathbf{c}}), \mathcal{T}(\hat{\mathbf{s}}), \mathcal{T}(\hat{\mathbf{a}})], \quad Q_c^t = \hat{f}, \quad Q_c^s = \hat{f}, \tag{2}$$

where $\mathcal{T}$ denotes random 3D transformations such as flipping, rotation, scaling, and translation, and $t$ and $s$ represent the teacher and student models, respectively.

The reconstructed queries $(Q_p, Q_c, \mathcal{R})$ and the original queries $Q_o$ are passed through the transformer decoder D of both teacher and student models to get the query features of the reconstructed and original object queries:

$$\hat{f}_c^t, \hat{f}_o^t = \mathrm{D}^t([(Q_c^t, Q_p^t, \mathcal{R}^t), Q_o^t], F_e^t, M), \quad \hat{f}_c^s, \hat{f}_o^s = \mathrm{D}^s([(Q_c^s, Q_p^s, \mathcal{R}^s), Q_o^s], F_e^s, M), \tag{3}$$

where $F_e$ denotes the encoder features and $M$ is an attention mask to prevent information leakage.

By explicitly reconstructing 3D object queries from teacher pseudo-labels, our method guarantees correspondence between teacher and student query features to achieve stable consistency training. The consistency loss is computed as the mean squared error (MSE) between the reconstructed query features of teacher and student:

$$\mathcal{L}_{\text{consistency}}(\hat{f}_c^s, \hat{f}_c^t) = \frac{1}{N} \sum_{i=1}^{N} \|\hat{f}_{c,i}^s - \hat{f}_{c,i}^t\|_2^2, \tag{4}$$

where $N$ is the number of query features.

**Discussion.** We introduce a fundamentally *3D-specific* form of query consistency. By reconstructing both geometric (position, size, orientation) and semantic (content) components of the object query, our method goes beyond cross-view consistency in the 2D Semi-DETR and provides the first principled solution for applying consistency-based regularization to DETR-style 3D object detection.

## 3.3 HYBRID MATCHING STRATEGY (HMS)

The hybrid matching strategy degenerates from a pre-training stage to a semi-supervision stage with softmax refinement to ensure robustness to noisy supervision. In the pre-training stage, we adopt a one-to-many (o2m) matching strategy to mitigate the suppression of potential positive proposals caused by the limited amount of annotated 3D data. Concretely, we create $K$-times augmented ground truths $Y = \{y^1, y^2, \ldots, y^K\}$, where $y^1 = y^2 = \ldots = y^K = y$ to increase the number of positive samples available for student training. A bipartite matching is then established by minimizing the pair-wise matching cost $\mathcal{L}_{\text{match}}(\hat{y}_{\sigma(i)}, Y_i)$:

$$\hat{\sigma}_{o2m} = \arg\min_{\sigma} \sum_{i=1}^{N} \mathcal{L}_{\text{match}}(\hat{y}_{\sigma(i)}, Y_i). \tag{5}$$

The pre-trained student is subsequently used to initialize the teacher model.

Although the one-to-many strategy increases proposal diversity, it is ill-suited for semi-supervised learning where pseudo-labels from the teacher are noisy. Multiple predictions matched to the same noisy pseudo-label risk amplifying errors and destabilizing training. To suppress noise amplification, we *degenerate* the matching strategy to one-to-one (o2o) in the semi-supervised stage:

$$\hat{\sigma}_{o2o} = \arg\min_{\sigma} \sum_{i=1}^{N} \mathcal{L}_{\text{match}}(\hat{y}_{\sigma(i)}, y_i). \tag{6}$$

This degeneration from o2m (pre-training) to o2o (semi-supervised learning) is critical for stabilizing DETR in 3D. Unlike 2D Semi-DETR where hybrid matching suffices, the 3D setting demands this degeneration to prevent volumetric pseudo-label noise from propagating.

**Activation Function.** Standard DETR-based 3D detectors typically adopt sigmoid activation in the classification head to produce independent confidence scores for each class. These scores are used to filter pseudo-labels in semi-supervised settings. However, sigmoid does not enforce mutual exclusivity and often results in uniformly low scores across classes. As a result, we observe that valid pseudo-labels tend to be discarded or ambiguous negatives to be retained. We thus replace the sigmoid with a softmax activation to emphasize the most confident class and enforce exclusivity. This sharper scoring mechanism significantly improves pseudo-label reliability in 3D, where semantic ambiguity is high, and volumetric overlaps are difficult to verify. Despite the fact that 2D Semi-DETR formulation does not require this modification, it is essential in 3D to reduce false positives and maintain stable learning.

**Loss Functions.** We denote the detection loss as $\mathcal{L}_{\text{det}}$. The pre-training loss with o2m matching is:

$$\mathcal{L}_{\text{pre-train}}^{\text{o2m}}(\hat{y}, Y) = \mathcal{L}_{\text{det}}^{\text{o2m}}(\hat{y}_{\hat{\sigma}(i)}, Y_i) + \lambda \mathcal{L}_{\text{iou}}(\hat{v}_{\hat{\sigma}(i)}, v_i), \tag{7}$$

where $\lambda$ balances the loss terms.

In the semi-supervised stage, we compute both supervised and unsupervised losses with o2o matching:

$$\mathcal{L}_{\text{sup}}^{\text{o2o}}(\hat{y}, y) = \mathcal{L}_{\text{det}}^{\text{o2o}}(\hat{y}_{\hat{\sigma}(i)}, y_i) + \lambda \mathcal{L}_{\text{iou}}(\hat{v}_{\hat{\sigma}(i)}, v_i), \quad \mathcal{L}_{\text{unsup}}^{\text{o2o}}(\hat{y}, y^{pseudo}) = \mathcal{L}_{\text{det}}^{\text{o2o}}(\hat{y}_{\hat{\sigma}(i)}, y_i^{pseudo}). \tag{8}$$

Finally, the total semi-supervised training loss is:

$$\mathcal{L}_{\text{semi-sup}} = \mathcal{L}_{\text{sup}}^{\text{o2o}} + \mathcal{L}_{\text{unsup}}^{\text{o2o}} + \mathcal{L}_{\text{consistency}}. \tag{9}$$

**Discussion.** Our training scheme introduces two novel 3D-specific adjustments absent in Semi-DETR: i) A *matching degeneration strategy* that transitions from o2m in pre-training to o2o in semi-supervision that is explicitly designed to combat noise amplification in 3D pseudo-labels. ii) Replacing the sigmoid with softmax activation to *enforce class exclusivity*, which leads to the improvement of pseudo-label filtering under high semantic ambiguity. These components provide a critical foundation for adapting DETR to semi-supervised 3D detection.

## 4 EXPERIMENTS

**Datasets.** We evaluate our approach on two benchmarks: ScanNet Dai et al. (2017) and SUN RGB-D Song et al. (2015), employing evaluation protocols from existing semi-supervised 3D object detection literature. ScanNet is a widely used dataset for 3D indoor scene with 1,201 training scenes and 312 validation scenes. We focus on the 18 semantic classes to be with prior studies. SUN RGB-D consists of 5,285 training scenes and 5,050 validation scenes. We evaluate the performance on 10 most common classes for comparison with previous methods.

**Evaluation Metrics.** We split the two datasets with various ratios of labeled data and unlabeled data. We allocate 5%, 10%, 20%, and 100% of labeled data for the ScanNet evaluation, and 5%, 10%, and 20% for SUN RGB-D evaluation. We report the standard mean Average Precision (mAP) under different IoU thresholds: mAP@0.25 for 0.25 IoU threshold and mAP@0.5 for 0.5 IoU threshold.

**Implementation Details.** We adopt V-DETR Shen et al. (2023) as the baseline, and the teacher and student models share the same network architecture. Following V-DETR, we use the AdamW optimizer Loshchilov (2017) with a base learning rate of 7e-4, and a weight decay of 0.1. The learning rate is warmed-up for 9 epochs, and then decreased to 1e-6 using the cosine schedule throughout the training process. For the pre-training stage, we train 2,040 epochs for convergence with the batch size 2. For the semi-supervised learning stage, we train 2,040 epochs with 4 labeled samples and 6 unlabeled samples per batch. The weights of the original detection loss remain the same as V-DETR, and the weight $\lambda = 10$.

### 4.1 COMPARISON WITH STATE-OF-THE-ART METHODS

The results on **ScanNet** in Table 1 and **SUN RGB-D** in Table 2 provide a comprehensive comparison across voting-based, voxel-based, and DETR-based semi-supervised 3D detectors. We make the following analysis of the results:

**Comparison with voting-based methods.** Our Semi-3DETR consistently outperforms all voting-based semi-supervised methods, including SESS Zhao et al. (2020), 3DIoUMatch Wang et al. (2021),

Table 1: Comparisons of semi-supervised 3D object detection methods on ScanNet val set with 5%, 10%, 20%, and 100% labeled data. Best results are in **bold**, second best are underlined. Gains are computed against the corresponding fully supervised (FS) baseline: VoteNet, FCAF3D, or V-DETR. Diff3DETR is shown with dual gains over VoteNet and V-DETR since it mixes paradigms.

| Type | Model | 5% | | 10% | | 20% | | 100% | |
|---|---|---|---|---|---|---|---|---|---|
| | | mAP@0.25 | mAP@0.5 | mAP@0.25 | mAP@0.5 | mAP@0.25 | mAP@0.5 | mAP@0.25 | mAP@0.5 |
| FS Voting | VoteNet Qi et al. (2019) | 27.9 | 10.8 | 36.9 | 18.2 | 46.9 | 27.5 | 57.8 | 36.0 |
| FS Voxel | FCAF3D Rukhovich et al. (2022) | 43.8 | 29.3 | 51.1 | 35.7 | 58.2 | 42.1 | 69.5 | 55.1 |
| FS DETR | V-DETR Shen et al. (2023) | 45.2 | 31.7 | 54.1 | 39.2 | 60.1 | 44.2 | 69.7 | 55.1 |
| Voting | SESS Zhao et al. (2020) | 32.0 (+4.1) | 14.4 (+3.6) | 39.5 (+2.6) | 19.8 (+1.6) | 49.6 (+2.7) | 29.0 (+1.5) | 61.3 (+3.5) | 39.0 (+3.0) |
| Voting | 3DIoUMatch Wang et al. (2021) | 40.0 (+12.1) | 22.5 (+11.7) | 47.2 (+10.3) | 28.3 (+10.1) | 52.8 (+5.9) | 35.2 (+7.7) | 62.9 (+5.1) | 42.1 (+6.1) |
| Voting | NESIE Wang et al. (2023) | 40.5 (+12.6) | 23.8 (+13.0) | 48.8 (+11.9) | 31.1 (+12.9) | 54.5 (+7.6) | 37.3 (+9.8) | 63.8 (+6.0) | 44.1 (+8.1) |
| Voting | Diffusion-SS3D Ho et al. (2023) | 43.5 (+15.6) | 27.9 (+17.1) | 50.3 (+13.4) | 33.1 (+14.9) | 55.6 (+8.7) | 36.9 (+9.4) | 64.1 (+6.3) | 43.2 (+7.2) |
| Voting/Transformer | Diff3DETR Deng et al. (2025) | 45.1 (+17.2) / (-0.1) | 29.2 (+18.4) / (-2.5) | 51.6 (+14.7) / (-2.5) | 34.2 (+16.0) / (-5.0) | 57.0 (+10.1) / (-3.1) | 38.2 (+10.7) / (-6.0) | 65.7 (+7.9) / (-4.0) | 44.9 (+8.9) / (-10.2) |
| Voxel | DQS3D Gao et al. (2023) | 49.2 (+5.4) | 35.0 (+5.7) | 57.1 (+6.0) | 41.8 (+6.1) | 64.3 (+6.1) | 48.5 (+6.4) | 71.9 (+2.4) | 56.3 (+1.2) |
| DETR | Ours (Semi-3DETR) | 54.1 (+8.9) | 39.5 (+7.8) | 58.2 (+4.1) | 43.5 (+4.3) | 63.2 (+3.1) | 47.2 (+3.0) | 72.0 (+2.3) | 57.6 (+2.5) |

Table 2: Comparisons of semi-supervised 3D object detection methods on SUN RGB-D val set with 5%, 10%, and 20% labeled data. Best results are in **bold**, second best are underlined. Gains are computed against fully supervised (FS) backbones: VoteNet, FCAF3D, V-DETR. Diff3DETR is shown with dual gains over VoteNet and V-DETR since it mixes paradigms.

| Type | Model | 5% | | 10% | | 20% | |
|---|---|---|---|---|---|---|---|
| | | mAP@0.25 | mAP@0.5 | mAP@0.25 | mAP@0.5 | mAP@0.25 | mAP@0.5 |
| FS Voting | VoteNet Qi et al. (2019) | 29.9 | 10.5 | 38.9 | 17.2 | 45.7 | 22.5 |
| FS Voxel | FCAF3D Rukhovich et al. (2022) | 49.5 | 31.7 | 50.7 | 33.4 | 54.3 | 36.5 |
| FS DETR | V-DETR Shen et al. (2023) | 41.2 | 22.7 | 49.1 | 30.3 | 53.7 | 34.2 |
| Voting | SESS Zhao et al. (2020) | 34.2 (+4.3) | 13.1 (+2.6) | 42.1 (+3.2) | 20.9 (+3.7) | 47.1 (+1.4) | 24.5 (+2.0) |
| Voting | 3DIoUMatch Wang et al. (2021) | 39.0 (+9.1) | 21.1 (+10.6) | 45.5 (+6.6) | 28.8 (+11.6) | 49.7 (+4.0) | 30.9 (+8.4) |
| Voting | NESIE Wang et al. (2023) | 41.1 (+11.2) | 21.8 (+11.3) | 47.4 (+8.5) | 29.2 (+12.0) | 53.4 (+7.7) | 31.2 (+8.7) |
| Voting | Diffusion-SS3D Ho et al. (2023) | 43.9 (+14.0) | 24.9 (+14.4) | 49.1 (+10.2) | 30.4 (+13.2) | 51.4 (+5.7) | 32.4 (+9.9) |
| Voting/Transformer | Diff3DETR Deng et al. (2025) | 45.7 (+15.8) / (+4.5) | 26.2 (+15.7) / (+3.5) | 50.2 (+11.3) / (+1.1) | 31.7 (+14.5) / (+1.4) | 53.2 (+7.5) / (-0.5) | 34.0 (+11.5) / (-0.2) |
| Voxel | DQS3D Gao et al. (2023) | 53.2 (+3.7) | 35.6 (+3.9) | 55.7 (+5.0) | 38.2 (+4.8) | 58.0 (+3.7) | 42.3 (+5.8) |
| DETR | Ours (Semi-3DETR) | 46.0 (+4.8) | 28.3 (+5.6) | 52.5 (+3.4) | 34.1 (+3.8) | 55.6 (+1.9) | 36.9 (+2.7) |

NESIE Wang et al. (2023), and Diffusion-SS3D Ho et al. (2023). This holds across all label ratios, which highlights the advantage of our DETR-based formulation over frameworks that rely on hand-crafted priors such as centroid voting and local feature grouping.

**Comparison with voxel-based DQS3D.** On **ScanNet**, our Semi-3DETR surpasses DQS3D Gao et al. (2023). This confirms the strength of point-based reasoning in cluttered and sparse 3D environments where voxel quantization introduces artifacts. On **SUN RGB-D**, our Semi-3DETR slightly underperforms compared to DQS3D, which benefits from structured indoor layouts that favor voxelized representations.

**Gain-based evaluation.** As shown in the first three rows of each table, the fully supervised backbones: VoteNet Qi et al. (2019), FCAF3D Rukhovich et al. (2022) and V-DETR Shen et al. (2023) differ substantially in absolute performance. This makes direct cross-category comparisons unfair. A more fair assessment is obtained by comparing the *gains of semi-supervised methods over their fully supervised backbones*. This normalizes the strength of the baseline and reveals how effectively each paradigm benefits from unlabeled data.

**Insights across paradigms.** Voting-based methods have been the most extensively explored and indeed show large gains over VoteNet, *e.g.* +10–13 mAP points at 5% labels. However, their absolute performance remains behind the voxel- and DETR-based frameworks. Although Voxel-based DQS3D achieves strong absolute results, its gains over FCAF3D are comparable to those of our Semi-3DETR over V-DETR. This indicates that backbone strength plays an important role. In contrast, our Semi-3DETR achieves consistent improvements over V-DETR across all label ratios. This shows that DETR can be effectively adapted to semi-supervised 3D detection despite its unique challenges of query correspondence, volumetric IoU sensitivity, and matching instability.

**Clarifying Diff3DETR.** We report dual gains relative to both voting- and DETR-based backbones for Diff3DETR Deng et al. (2025), *i.e.* VoteNet and V-DETR, respectively. Its large improvements over VoteNet confirm that it is fundamentally a *voting-based method with a transformer backbone* rather than a true DETR framework. Its weaker performance relative to our Semi-3DETR under DETR baselines further underscores the necessity of our approach: a dedicated DETR-specific design for semi-supervised 3D detection.

In summary, although voting- and voxel-based semi-supervised methods have been well studied, our Semi-3DETR is the first to open a new research direction in DETR-based semi-supervised 3D detection. Our proposed method achieves competitive gains and establishes a complementary paradigm to the voxel-based approach DQS3D.

Table 3: Ablation study of each component in our proposed method.

| | IoU estimation | Query consistency | 3D Semi-supervised learning-aware adjustments | | ScanNet 5% | | ScanNet 10% | |
|---|---|---|---|---|---|---|---|---|
| | | | Matching degeneration | Softmax activation | mAP@0.25 | mAP@0.5 | mAP@0.25 | mAP@0.5 |
| [A] | - | - | - | - | 45.2 | 31.7 | 54.1 | 39.2 |
| [B] | | | | | 11.7 | 8.2 | 13.7 | 7.9 |
| [C] | ✓ | | | ✓ | 51.5 | 34.0 | 55.3 | 41.5 |
| [D] | ✓ | ✓ | | ✓ | 52.7 | 35.9 | 57.0 | 42.6 |
| [E] | ✓ | ✓ | ✓ | | 13.4 | 11.4 | 16.1 | 12.0 |
| [F] | ✓ | ✓ | ✓ | ✓ | **54.1** | **39.5** | **58.2** | **43.5** |

## 4.2 ABLATION STUDY AND ANALYSIS

**Effectiveness of Each Component.** We ablate the three key components of our Semi-3DETR: the *query feature-driven IoU estimation head*, the *query consistency regularization*, and the *3D semi-supervised learning-aware adjustments* (matching degeneration and softmax activation). Table 3 summarizes the results.

[A] corresponds to the supervised-only baseline trained on labeled data. [B] shows that naive introduction of unlabeled data with pseudo-labels severely degrades performance. This confirms the need for robust filtering and regularization mechanisms.

Table 4: Ablation of the query consistency components.

| Attn_mask | Pseudo-labels w filtering | Pseudo-labels w/o filtering | ScanNet 5% | |
|---|---|---|---|---|
| | | | mAP@0.25 | mAP@0.5 |
| ✓ | ✓ | | 53.5 | 38.1 |
| ✓ | | ✓ | **54.1** | **39.5** |
| | | ✓ | 48.0 | 31.7 |

Adding the ***IoU estimation head*** ([C]) substantially mitigates this issue by filtering out low-quality and duplicated pseudo-labels, yielding a large gain over [B]. Incorporating ***query consistency regularization*** ([D]) further provides another clear improvement. This demonstrates its role in enforcing semantic invariance under weak–strong augmentations and stabilizing student training. Compared to [A], [D] improves by +7.5 mAP@0.25 and +4.2 mAP@0.5 at 5% labels, which validate the effectiveness of these two components together.

The ***matching degeneration strategy*** ([D] → [F]) further boosts performance, particularly at higher IoU thresholds. This confirms its ability to alleviate noise amplification and preserve positives by balancing one-to-many and one-to-one matching.

Finally, replacing ***softmax with sigmoid activation*** ([E]) causes a severe drop. This agrees with our motivation: softmax enforces mutual exclusivity and sharpens class confidence to make pseudo-label filtering more reliable. In contrast, sigmoid treats classes independently and often assigns low and ambiguous scores across all categories.

Table 5: Effects of different IoU module.

| Method | ScanNet 5% | |
|---|---|---|
| | mAP@0.25 | mAP@0.5 |
| IoU module of 3DIoUMatch Wang et al. (2021) | 50.7 | 34.5 |
| Ours | **54.1** | **39.5** |

Overall, the full model [F] achieves the best results. This confirms that each component contributes synergistically to address the unique challenges of semi-supervised DETR-based 3D detection: filtering volumetric pseudo-labels, enforcing query-level consistency, and stabilizing matching under noisy supervision. The total gain of **+8.9 mAP@0.25** and **+7.8 mAP@0.5** over the supervised baseline at 5% ScanNet demonstrates their collective effectiveness of these components.

**Effectiveness of Query Consistency Components.** Table 4 analyzes the role of each component in our query consistency-based regularization scheme. Row 1 shows that using filtered pseudo-labels with consistency regularization already yields gains by ensuring that supervision is derived from reliable proposals. Interestingly, Row 2 shows that extending regularization to *all* pseudo-labels (w/o filtering) achieves the best overall results. Although unfiltered pseudo-labels inevitably

Table 6: Vary pseudo-label threshold $\tau$.

| $\tau$ | ScanNet 5% | |
|---|---|---|
| | mAP@0.25 | mAP@0.5 |
| 0.2 | 48.9 | 33.3 |
| 0.3 | 51.8 | 36.7 |
| 0.4 | **54.1** | **39.5** |
| 0.5 | 50.0 | 37.7 |
| 0.6 | 45.8 | 30.5 |

include some noisy and duplicate proposals, they also provide broader scene coverage. This wider spatial supervision improves the ability of the model to learn semantic feature invariance to outweigh the negative effects of noise. Finally, Row 3 demonstrates the importance of the attention mask: removing it causes a sharp performance drop. This confirms our hypothesis that the attention mask is crucial to prevent information leakage between reconstructed and original queries, thereby maintaining the integrity of consistency learning. These results validate our design choices: pseudo-label

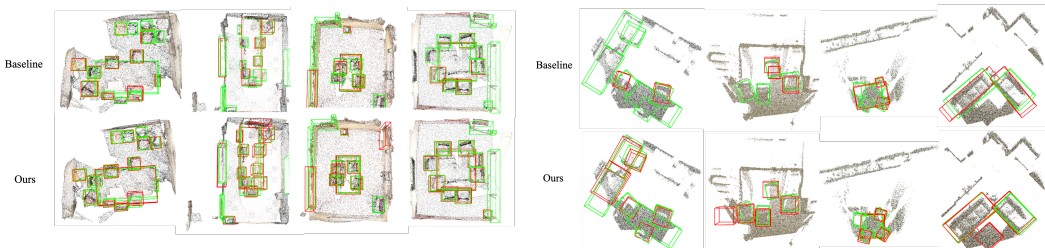

Figure 2: Qualitative comparison of baseline and our work on ScanNet val set with 5% labeled data.

Figure 3: Qualitative comparison of baseline and our work on SUN RGB-D val set with 10% labeled data.

diversity helps build invariance, but only when paired with strict masking to avoid trivial information sharing.

**Analysis of Other Components.** Table 5 compares our IoU estimation head with the IoU estimation module in 3DIoUMatch Wang et al. (2021). Our approach consistently achieves higher performance. This highlights the importance of tailoring IoU estimation specifically for DETR-based semi-supervised 3D detection. Unlike voxel- or voting-based detectors, DETR requires careful filtering of pseudo-labels to avoid propagation of duplicates and spatially inconsistent proposals. Our IoU estimation head effectively addresses this by combining query features with volumetric overlap to enable more reliable supervision. Furthermore, the threshold $\tau$ plays a crucial role as a confidence-based filter that balances the precision–recall trade-off of pseudo-labels. As shown in Table 6, performance peaks at $\tau = 0.4$. A lower threshold destabilizing training by admission of excessive noise, while a higher threshold reduces supervision signal by discarding too many pseudo-labels. This validates our design choice of IoU-guided filtering, which ensures stable learning while preserving sufficient pseudo-label coverage for effective semi-supervised training.

## 5 QUALITATIVE RESULTS

Figures 2 and 3 present qualitative comparisons between the baseline and our Semi-3DETR on ScanNet with 5% labeled data and SUN RGB-D with 10% labeled data, respectively. The ground truth boxes are shown in green and the predictions in red.

Across both datasets, our method consistently produces more complete and accurate detections than the baseline trained only with labeled data. For example, the baseline detects only 6 objects while our Semi-3DETR correctly identifies 10 in the ScanNet scene in the last column of Figure 2 that contains 13 objects of interest. This shows that our framework can leverage unlabeled data to substantially improve recall without sacrificing precision.

These qualitative results align with our motivation: While existing semi-supervised methods for voting- or voxel-based detectors rely on strong architectural priors, our Semi-3DETR effectively extends the prior-free DETR paradigm to semi-supervised 3D detection. This enables robust predictions even with scarce annotations.

## 6 CONCLUSION

We introduced Semi-3DETR, the first semi-supervised framework for DETR-based 3D object detection. Our Semi-3DETR tackles key challenges of pseudo-label noise, query correspondence, and matching instability through a query feature-driven IoU head, query consistency regularization, and a matching degeneration strategy with softmax refinement. Experiments on ScanNet and SUN RGB-D validate its effectiveness. Specifically, our Semi-3DETR shows consistent gains over supervised DETR baselines and competitive performance with voxel-based approaches. These results underscore the role of our approach as a complementary label-efficient alternative. Although voxel-based methods such as DQS3D still report strong results in absolute performance, they rely on discretization and are less flexible than point-based DETR. Our work opens new directions for exploring semi-supervised DETR in 3D. Future efforts will focus on scaling to larger outdoor datasets that enhance efficiency with sparse or hybrid representations, and leveraging multi-modal signals to further improve robustness under limited supervision.

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
