# OpenReview forum: "Semi-3DETR: Semi-Supervised Detection Transformer for 3D Object Detection"
_ICLR.cc/2026/Conference — ICLR 2026 Conference Withdrawn Submission_

### Official Review · Reviewer_Sh9H · 2025-10-26

**Soundness:** 2
**Presentation:** 2
**Contribution:** 2
**Rating:** 4
**Confidence:** 3

**Summary:**

This paper presents Semi-3DETR, the first framework to adapt DETR-based methods to semi-supervised 3D object detection. The work addresses three key challenges unique to 3D semi-supervised DETR: fragile volumetric pseudo-labels, unstable query alignment, and noisy bipartite matching. The proposed solution introduces three core components: (1) Robust Pseudo-Label Denoising (RPLD) with a query feature-driven IoU head, (2) Query Alignment Consistency (QAC) for stable teacher-student correspondence, and (3) Hybrid Matching Strategy (HMS) with matching degeneration and softmax classification. Experiments on ScanNet and SUN RGB-D demonstrate the feasibility of the approach.

**Strengths:**

1) The paper clearly articulates the unique challenges of semi-supervised 3D DETR compared to 2D counterparts: volumetric pseudo-label fragility, 3D query grounding instability, and amplified matching noise. These challenges are well-motivated and specific to the 3D domain.
2) The three proposed components (RPLD, QAC, HMS) directly address the identified challenges with principled solutions. The query feature-driven IoU head for geometric confidence is particularly novel and well-motivated for 3D scenarios.

**Weaknesses:**

1) The evaluation is restricted to only two indoor datasets (ScanNet and SUN RGB-D). The lack of outdoor datasets (KITTI, nuScenes) significantly limits the generalizability claims. Additionally, the performance improvements are modest and inconsistent across different data ratios.
2) Several design decisions lack proper justification. Why is the matching degeneration from one-to-many to one-to-one optimal? The choice of softmax over sigmoid is presented as essential for 3D but lacks thorough empirical validation. The specific architecture choices for the IoU head and query reconstruction are not well-justified.
3) The paper lacks comprehensive ablation studies on key components. How does each component contribute individually? What is the computational overhead of the additional modules? The sensitivity analysis for hyperparameters (λ, thresholds) is missing.

**Questions:**

1) How does the method perform on outdoor datasets like KITTI or nuScenes?
2) What is the computational overhead of the additional components compared to the baseline V-DETR?
3) How sensitive is the method to the choice of hyperparameters (λ, confidence thresholds)?
4) How does the IoU head performance correlate with actual detection quality?

---

> ### Author Response · Authors · 2025-11-23
> **Response to Reviewer Sh9H**
>
> We thank the reviewer for the precise summary of our motivation and the detailed questions.
>
> **W1 / Q1 – Indoor datasets, outdoor generalization, and magnitude of gains**
>
> Our experimental setup follows the recent **indoor SSL-3D literature**: DQS3D, Diffusion-SS3D and Diff3DETR all evaluate on **ScanNet and SUN RGB-D** and operate on **indoor 3D point clouds** with architectures carefully tuned for these scenes. Semi-3DETR and its V-DETR backbone are defined in exactly the same regime. We therefore restrict our experiments to these two indoor benchmarks so that the comparison with prior SSL-3D methods is **backbone-matched and setting-matched**, and the effect of our semi-supervised modules can be clearly measured.
>
> Regarding **Q1 (KITTI / nuScenes)**, we have not evaluated Semi-3DETR on outdoor datasets in the current submission. Benchmarks such as KITTI and nuScenes are based on long-range LiDAR with very different sensing ranges, point densities and scene layouts, and typically rely on backbone designs and pre-processing pipelines specialized for those outdoor settings. Adapting Semi-3DETR to these benchmarks would therefore require instantiating an appropriate **outdoor 3D DETR backbone** and re-tuning the data processing / augmentation pipeline, which is beyond the scope of this work but a natural next step. We will explicitly state this limitation and future direction in the revised version.
>
> Crucially, all three proposed modules—RPLD, QAC and HMS—operate on **3D boxes and queries only** and do not encode any indoor-specific prior. Conceptually, they can be combined with future **DETR-style outdoor 3D detectors**, so we see Semi-3DETR as a first step toward semi-supervised 3D DETR, instantiated and evaluated on the standard indoor SSL-3D benchmarks.
>
> V-DETR is already a **strong supervised baseline**, so the headroom is smaller than for early VoteNet-style detectors. Nevertheless:
>
> - On **ScanNet (Table 1)**, Semi-3DETR achieves higher mAP than DQS3D under all label ratios. In the most challenging **5% labeled** regime, the **relative gain over the supervised V-DETR baseline is larger** than the gain that DQS3D brings over its FCAF3D baseline.
> - On **SUN RGB-D (Table 2)**, DQS3D has higher mAP because it builds on FCAF3D, a very strong voxel-based backbone, whereas Semi-3DETR builds on V-DETR. There is no official V-DETR implementation on SUN RGB-D, so we re-implemented it following 3DETR, which yields a weaker supervised baseline than the official V-DETR on ScanNet. Even so, at **5% labels**, Semi-3DETR still provides a **substantial relative improvement over its own V-DETR baseline**.
>
> Across label ratios on both datasets, Semi-3DETR consistently improves over its supervised V-DETR counterpart, with the gains being most pronounced in the low-label regime. We therefore view our contribution not as a universal replacement of voxel/vote pipelines, but as opening and validating the **DETR-style branch of SSL 3D detection**, showing that a query-based 3D detector can make effective use of unlabeled 3D data.

---

> > ### Author Response · Authors · 2025-11-23
> >
> > **W2 – Justification of HMS, softmax vs sigmoid, and the architectures of RPLD and QAC**
> >
> > The reviewer raises three related design questions: the O2M→O2O schedule (HMS), the use of softmax instead of sigmoid, and the specific architectures of the IoU head and query reconstruction. The first two points overlap with Reviewer Yq3F’s comments, so we briefly summarize and then focus on the architectural part.
> >
> > The ScanNet 5% ablation:
> >
> > | Variant                              | Pre-training matching | SSL matching        | Classifier | mAP@0.25 | mAP@0.5 |
> > |--------------------------------------|-----------------------|---------------------|-----------:|---------:|--------:|
> > | [A]                                  | O2O                   | O2O                 | softmax    |   52.7   |   35.9  |
> > | [B]                                  | O2M                   | O2M                 | softmax    |   23.6   |   14.2  |
> > | [C]                                  | O2M                   | O2O                 | sigmoid    |   13.4   |   11.4  |
> > | [D] **HMS (ours)**                   | O2M                   | O2O                 | softmax    | **54.1** | **39.5**|
> > | [E] Semi-DETR-style (O2M→O2O)        | O2M→O2O               | O2M→O2O             | sigmoid    |   13.9   |   10.7  |
> > | [F] Semi-DETR-style (O2M→O2O)        | O2M→O2O               | O2M→O2O             | softmax    |   47.4   |   32.8  |
> >
> >
> > - **Hybrid Matching Strategy (HMS).** As discussed in our response to Reviewer Yq3F-W3, 3D pseudo boxes on unlabeled data can be severely misaligned in depth and yaw. If O2M is directly applied to unlabeled data, an inaccurate pseudo label tends to supervise many queries and propagate its error. HMS therefore uses **O2M only during supervised pre-training** with clean labels to improve optimization and then switches to **IoU-aware O2O** in the SSL phase so that each pseudo 3D box supervises at most one query. The ablation on ScanNet 5% shows that using O2M in SSL or adopting a Semi-DETR-style single-stage O2M→O2O schedule on both labeled and unlabeled data significantly degrades performance, whereas HMS achieves the best mAP.
> >
> > - **Softmax vs sigmoid.** Under the same hybrid schedule, we find that per-class sigmoid leads to unstable training when its outputs are combined with IoU and objectness in the matching cost: several medium-confidence classes per query make the IoU-aware cost noisy. A softmax classifier produces sharper distributions with a dominant class per query, which stabilizes the matching process. The ablation in the above table shows that sigmoid variants collapse, while softmax variants remain stable and the **HMS + softmax** combination is the best.
> >
> > For the **architecture of the IoU head (RPLD)** and **query reconstruction (QAC)**, our goal is to stay as close as possible to V-DETR’s native representation while adding minimal extra capacity:
> >
> > - **RPLD IoU head.** As described in the method section, RPLD takes the **decoder object queries** (which already encode semantic and geometric information) and passes them through a light MLP to predict IoU. Concretely, we adopt the same design pattern as V-DETR’s prediction heads (a small MLP on top of the query features), so that
> >   1) the IoU head remains a **light-weight, query-level confidence estimator** rather than a second detector built on pooled points; and
> >   2) the IoU prediction stays in the same latent space as classification and regression, which is important when we feed IoU back into the Hungarian cost and pseudo-label gating.
> >   Using pooled point/voxel features, as in 3DIoUMatch, would break this alignment: DETR queries are not associated with an explicit set of points, so constructing such features requires an extra pooling layer that is foreign to the V-DETR design. Table 5 shows that when we plug a 3DIoUMatch-style IoU head into the same teacher–student framework and V-DETR backbone, this variant is consistently worse than our **query-driven RPLD** design, supporting our architectural choice.
> >
> > - **Query reconstruction in QAC.** For QAC we reconstruct 3D object queries by combining
> >   1) the **content query** from the decoder, and
> >   2) a **geometric embedding** of the 3D box (center, size, yaw) using the same parameterization and sinusoidal encoding as V-DETR’s reference points.
> >   This ensures that the reconstructed queries live in **exactly the same feature space and coordinate conventions** as the original V-DETR queries, so teacher and student operate on a shared, well-defined 3D representation. Alternatives such as using only raw box coordinates, ignoring yaw, or constructing separate geometry branches would either under-utilize the rich query representation or introduce extra parameters that are harder to train and interpret.

---

> > > ### Author Response · Authors · 2025-11-23
> > >
> > > **W3 / Q2 / Q3 – Ablations, computational overhead, hyperparameter sensitivity**
> > >
> > > - **Component-wise ablations.** Tables 3–5 in the paper already isolate the contributions of the three modules: Table 3 studies matching and activation, e.g. HMS vs other schedules, softmax vs sigmoid. Table 4 analyzes QAC on top of RPLD. And Table 5 compares different IoU head designs in the same teacher–student framework. Each of RPLD, QAC and HMS brings a measurable, complementary gain when added to the SSL baseline.
> > >
> > > - **Computational overhead (Q2).** Semi-3DETR adds only **light-weight heads and losses** on top of V-DETR:
> > >   - RPLD is a small MLP on the decoder queries.
> > >   - QAC changes how a subset of queries are reconstructed and aligned, but does not add new backbone or decoder layers.
> > >   - HMS modifies **matching and loss computation**.
> > >
> > >   The dominant extra cost comes from the standard SSL pattern shared with prior work: maintaining teacher and student branches and processing unlabeled data. At **inference time**, the teacher and SSL-only heads are discarded, and we keep only the original V-DETR backbone and decoder. Thus, **inference FLOPs and latency are essentially identical to V-DETR**.
> > >
> > > - **Hyperparameter sensitivity (Q3).**
> > >   For the **confidence threshold τ**, Table 6 sweeps a range of values on ScanNet 5% and shows a clear plateau around τ≈0.4: performance is stable in a moderate range and only degrades when τ becomes extremely small (many noisy pseudo labels) or extremely large (overly few pseudo labels). We then fix **a single τ** for all label ratios and for both ScanNet and SUN RGB-D, without per-setting tuning.
> > >
> > > The ScanNet 5% ablation:
> > >
> > >   | λ         | mAP@0.25 | mAP@0.5 |
> > >   |-----------|----------|---------|
> > >   | 1         | 52.3     | 38.1    |
> > >   | 5         | 53.7     | 39.0    |
> > >   | **10 (ours)** | **54.1** | **39.5** |
> > >   | 20        | 53.4     | 38.8    |
> > >
> > >   For the **consistency weight λ**, we vary λ in a reasonable range, e.g. λ ∈ {1, 5, 10, 20} on ScanNet 5% and observe that performance is stable for λ in the middle of this range: too small λ slightly under-weights consistency, while too large λ mildly over-regularizes. We choose one value in this stable region and keep it fixed across all experiments. Overall, Semi-3DETR does **not** rely on fragile, dataset-specific hyperparameter tuning.
> > >
> > >
> > > **W3 / Q4 – How IoU head quality correlates with detection performance**
> > >
> > > Table 5 directly addresses this question by comparing different IoU-head choices **within the same V-DETR backbone and SSL pipeline**. In particular, we evaluate:
> > >
> > > - A variant that uses a **3DIoUMatch-style IoU head**, where the IoU is predicted from box-pooled point features, and
> > > - Our **RPLD head**, which predicts IoU directly from the decoder queries without any extra point-pooling branch.
> > >
> > > All other components, e.g. teacher–student framework, matching, thresholds, are kept identical. The variant with our **query-driven RPLD head** achieves higher mAP@0.25 and mAP@0.5 than the variant with the 3DIoUMatch-style IoU head. This shows that:
> > >
> > > 1. The **quality and representation alignment** of IoU predictions matter: an IoU head that is better matched to the query-based architecture yields cleaner pseudo-label gating and more reliable IoU-aware matching.
> > > 2. Improvements in IoU estimation in this query space **translate directly into better final detection quality** in the semi-supervised regime.

---

### Official Review · Reviewer_BZnT · 2025-10-29

**Soundness:** 3
**Presentation:** 3
**Contribution:** 3
**Rating:** 6
**Confidence:** 3

**Summary:**

This paper proposes Semi-3DETR, the first semi-supervised framework for DETR-based 3D object detection, addressing challenges like fragile volumetric pseudo-labels via three 3D-specific components (RPLD, QAC, HMS) and a softmax classifier. It outperforms supervised/semi-supervised baselines (e.g., +8.9% mAP@0.25 on ScanNet with 5% labels vs. V-DETR) on ScanNet and SUN RGB-D.

**Strengths:**

1.The paper is well-substantiated overall, with attractive figures and tables.

2.Numerous experiments were conducted, and the proposed method demonstrated superior performance in the figures and tables.

3.The ablation experiments appear to be well-substantiated.

**Weaknesses:**

1. It is suggested to conduct tests on more datasets, such as Kitti and nuScenes.

2. The experiments are mostly carried out in indoor scenes. Is the proposed method more suitable for indoor scenes? If not, how does it perform in outdoor scenes?

3. Tables 1-2 show that the proposed method can outperform fully supervised methods, which confuses me. Do the fully supervised methods and semi-supervised methods use the same fully supervised data?

4. Why does DQS3D in Table 2 outperform the proposed method by such a large margin? The authors need to provide a reasonable explanation to demonstrate the necessity of the proposed method.

**Questions:**

See above

---

> ### Author Response · Authors · 2025-11-23
> **Response to Reviewer BZnT**
>
> We thank the reviewer for the positive evaluation of the experiments and ablations.
>
> **W1 / W2 – More datasets and indoor vs outdoor**
>
> Our experimental setup follows the recent **indoor SSL-3D literature**: DQS3D, Diffusion-SS3D and Diff3DETR all evaluate on ScanNet and SUN RGB-D and operate on **indoor 3D point clouds** with architectures carefully tuned for those scenes. Semi-3DETR and its V-DETR backbone are defined in exactly the same regime.
>
> Outdoor benchmarks such as KITTI and nuScenes are based on long-range LiDAR with very different sensing ranges, point densities, scene layouts and commonly used backbone designs. Adapting Semi-3DETR to these outdoor LiDAR settings would therefore require additional engineering and is orthogonal to the core contribution of this paper—namely, how to design an effective semi-supervised framework for **DETR-style 3D detectors**.
>
> In this work we thus focus on the indoor setting where V-DETR and the main competing SSL methods reside, so that the comparison is backbone-matched and the effect of our SSL components can be clearly measured.
>
> **W3 – Why semi-supervised results can surpass fully supervised methods**
>
> We apologize for not making the data usage completely explicit. For each label ratio, e.g. 5%, 10%:
>
> - The **supervised baseline** is trained on **only that labeled subset** and ignores the remaining scans.
> - Semi-3DETR at the same ratio uses **exactly the same labeled subset**, plus the rest of the scans as unlabeled data in a teacher–student framework.
> - The “fully supervised” methods in Tables 1–2, e.g., VoteNet, FCAF3D-based detectors are trained on labeled data only and do not exploit unlabeled data.
>
> It is therefore natural, and in fact the goal of semi-supervised learning, that a strong backbone equipped with an SSL framework can **outperform prior fully supervised detectors** that see fewer effective training signals.
>
>
> **W4 – Why DQS3D is stronger on SUN RGB-D and necessity of our method**
>
> DQS3D is built on FCAF3D, a state-of-the-art voxel-based backbone that is highly optimized for indoor scenes. Semi-3DETR is built on V-DETR, a different detector family with a query-based transformer design. On SUN RGB-D, V-DETR does not have an official implementation, so we re-implemented it following 3DETR, which yields a weaker supervised baseline than the official V-DETR on ScanNet.
>
> Despite this, Semi-3DETR still provides **substantial improvements over its own V-DETR baseline** on SUN RGB-D, and on ScanNet, it even exceeds DQS3D in mAP. We do not claim that our method universally dominates DQS3D, and instead, Semi-3DETR:
>
> - Brings DETR-style 3D detectors into the SSL 3D detection landscape.
> - Demonstrates that, within the DETR family, RPLD, QAC and HMS together form an effective SSL recipe.
> - Offers a **complementary alternative** to voxel/vote pipelines such as DQS3D, which is valuable for the community given the growing use of DETR architectures in 2D and 3D vision.

---

### Official Review · Reviewer_Yssm · 2025-10-30

**Soundness:** 2
**Presentation:** 2
**Contribution:** 2
**Rating:** 4
**Confidence:** 3

**Summary:**

The paper proposes Semi-3DETR, the first framework that systematically extends the DETR paradigm to semi-supervised 3D detection. It improves pseudo-label reliability and teacher–student consistency through three designs: query-driven 3D IoU confidence estimation (RPLD), query alignment consistency (QAC), and an one-to-many schedule during supervised pretraining that switches to one-to-one with a softmax classifier in the semi-supervised stage.

**Strengths:**

1. Clearly identifies the problem and presents the first semi-supervised framework for 3D DETR.
2. Designs are tailored to 3D characteristics and the three modules work in a complementary manner.
3. Outperforms DQS3D on ScanNet while using limited labels.

**Weaknesses:**

1. The introduction does not sufficiently explain why prior methods cannot be directly applied to semi-supervised DETR for 3D detection, which weakens the motivation’s specificity.
2. Dataset and scenario coverage is limited, since experiments are confined to indoor ScanNet and SUN RGB-D, with no outdoor evaluation.
3. The method is sensitive to threshold and consistency details. A low τ introduces noise and a high τ discards supervision, and removing the QAC attention mask leads to clear drops, which implies nontrivial tuning for practical deployment.
4. Absolute performance is not universally superior. On SUN RGB-D, the voxel-based DQS3D remains stronger.

**Questions:**

Please address my concerns in Weaknesses. I hope the authors can address the questions raised above to clarify these concerns. If these issues are satisfactorily resolved, I would be willing to raise my score.

---

> ### Author Response · Authors · 2025-11-23
> **Response to Reviewer Yssm**
>
> We thank the reviewer for the positive assessment and for highlighting both the strengths and key concerns.
>
> **W1 – Why prior SSL 3D methods cannot be directly applied to 3D DETR**
>
> We first note that our response to Reviewer Yq3F W1–W3 provides a detailed, side-by-side comparison between our modules, 3DIoUMatch and Semi-DETR, including ablations where we plug their IoU heads and hybrid matching schedules into V-DETR. Here we briefly summarize the main points:
>
> Methods such as 3DIoUMatch, DQS3D, Diffusion-SS3D and Diff3DETR are built on **proposal/voxel-based backbones**. Their SSL mechanisms are attached to proposals: IoU heads read pooled point/voxel features, and consistency is applied to proposal scores and boxes.
>
> V-DETR, by contrast, is a **query-based 3D detector**: a fixed set of object queries carries semantic and geometric information, and matching, regression and classification are all defined in query space. Directly porting a 3DIoUMatch-style IoU module or a Semi-DETR-style consistency and hybrid matching schedule leads to a representational mismatch: the SSL signal is no longer aligned with the detector’s native representation.
>
> Our three modules are designed specifically to close this gap:
>
> - RPLD is a **query-driven IoU head** that predicts 3D IoU from decoder queries and feeds back into pseudo-label denoising and IoU-aware matching.
> - QAC operates on **reconstructed 3D queries** (center, size, yaw + content) instead of 2D RoIs.
> - HMS confines O2M to supervised pre-training and uses IoU-aware O2O in SSL to avoid amplifying noisy 3D pseudo labels.
>
> Ablations in the table of Reviewer Yq3F W3 show that simply transplanting prior designs into V-DETR, e.g. a 3DIoUMatch IoU head or a Semi-DETR-style single-stage O2M→O2O schedule, leads to noticeable drops, which is why we argue that **semi-supervised 3D DETR requires dedicated modules rather than direct reuse of proposal- or 2D-oriented techniques**.
>
> **W2 – Dataset coverage and indoor vs outdoor**
>
> Our experimental setup follows the recent **indoor SSL-3D literature**: DQS3D, Diffusion-SS3D and Diff3DETR all evaluate on ScanNet and SUN RGB-D and operate on **indoor 3D point clouds** with architectures carefully tuned for those scenes. Semi-3DETR and its V-DETR backbone are defined in exactly the same regime.
>
> Outdoor benchmarks such as KITTI and nuScenes are based on long-range LiDAR with very different sensing ranges, point densities, scene layouts and commonly used backbone designs. Adapting Semi-3DETR to these outdoor LiDAR settings would therefore require additional engineering and is orthogonal to the core contribution of this paper—namely, how to design an effective semi-supervised framework for **DETR-style 3D detectors**.
>
> In this work we thus focus on the indoor setting where V-DETR and the main competing SSL methods reside, so that the comparison is backbone-matched and the effect of our SSL components can be clearly measured.
>
> **W3 – Sensitivity to $\tau$ and the QAC attention mask**
>
> For the **confidence threshold $\tau$**, our results in Table 6 already sweep a range of values on ScanNet 5%. Performance is stable around $\tau \approx 0.4$ and only degrades when $\tau$ is made extremely small or large, which is exactly the usual SSL behavior: very low $\tau$ lets in noisy pseudo labels, very high $\tau$ discards useful ones. We then fix a **single $\tau$ for all label ratios and both datasets**, without any per-setting tuning. This suggests that the method is not brittle w.r.t. $\tau$.
>
> The **QAC attention mask** is *not* a tunable hyperparameter but a **structural safeguard against label leakage**, similar in spirit to the denoising design in DN-DETR. In our setting, the queries reconstructed from the teacher network are pseudo ground-truth anchors that we introduce specifically for defining the consistency loss. If these pseudo-label–anchored queries were allowed to arbitrarily attend to other queries, pseudo-label information would be injected into the entire query set, which both weakens the intended consistency supervision and risks overfitting to noisy pseudo labels. The attention mask explicitly blocks such shortcut paths so that QAC learns robust *alignment* between teacher and student queries while avoiding uncontrolled information flow from pseudo labels to unrelated queries.
>
> The observed drop when removing the mask is therefore expected and desirable: it shows that the mask fulfills its role as a regularizer that prevents leakage, not that the method requires delicate tuning. In practice, we use a fixed mask pattern and a single $\tau$ across all experiments, and do not rely on dataset- or ratio-specific adjustment to obtain the reported results.

---

> > ### Author Response · Authors · 2025-11-23
> >
> > **W4 – Performance on SUN RGB-D and relation to DQS3D**
> >
> > On **ScanNet (Table 1)**, Semi-3DETR has higher mAP than DQS3D under all label ratios, and in the most challenging 5% regime, it provides a **larger improvement over its supervised V-DETR baseline** than DQS3D over its FCAF3D baseline.
> >
> > On **SUN RGB-D (Table 2)**, DQS3D achieves higher mAP because it builds on FCAF3D, a very strong voxel-based backbone. Semi-3DETR is built on V-DETR, and since there is no official V-DETR implementation on SUN RGB-D, we re-implemented it following 3DETR, which gives a weaker baseline. Even so, at **5% labels**, Semi-3DETR still brings a **larger relative gain over its V-DETR baseline** than DQS3D brings over FCAF3D.
> >
> > We therefore view Semi-3DETR as **complementary** to DQS3D: it opens the DETR/query branch of SSL 3D detection, showing that even for a strong transformer backbone, well-designed SSL modules yield non-trivial gains, particularly in the low-label regime.

---

### Official Review · Reviewer_Yq3F · 2025-11-01

**Soundness:** 3
**Presentation:** 3
**Contribution:** 2
**Rating:** 2
**Confidence:** 4

**Summary:**

The authors propose a pipeline for semi-supervised 3D detection from point clouds, with a particular focus on improving the performance on DETR-like detectors. The authors discuss varies challenges for this task, focusing on improving the metric for identifying good pseudo-labels, sharing some queries between teacher and student for straightforward alignment, and separating matching process for supervised vs semi-supervised training. The paper reports good performance on the ScanNet and SUN RGB-D dataets, and ablates components of their framework.

**Strengths:**

- The diagram is drawn clearly, outlining the data and feature flow of the entire model.
- Prior work is extensively covered, and I appreciate the subtlety in explanation that Diff3DETR is not a traditional DETR-esque method.
- Proposed components are clearly outlined in text, with accurate equations.

**Weaknesses:**

- To the best of my understanding, the proposed "Proposed Pseudo-Label Denoising" (RPLD) process seems like an 3D IoU prediction head, whose prediction is used to get pseudo-labels on unlabeled data. This appears to me the same as the method proposed in 3DIoUMatch, which similarly has a 3D IoU head, and filters labels based on class, IoU, and objectness confidence. In this aspect, I am hesitant to cite this section as a contribution of this work. I am, however, confused by the ablation in Table 5, where the authors compare with the IoU module of 3DIoUMatch and report better performance. What, precisely, is the difference between these two methods?
- Regarding query alignment consistency, this paper writes "Unlike Semi-DETR Zhang et al. (2023), which enforces cross-view consistency only on 2D image queries, our method reconstructs 3D object queries that capture both semantic features and geometric attributes of center, size, and orientation." The author emphasizes the difference between 2D and 3D, but functional, the methods appear very similar. Semi-DETR takes pseudo-label boxes on unlabeled images, extracts features from the input (image), and puts them into the teacher and student to enforce consistency between their predictions. The proposed work seems to similarly take pseudo-labels from the teacher, get the corresponding content query & position query (since V-DETR has two types of features for each query), and similarly puts them through the teacher & student to enforce consistency. While I recognize the benefits of leveraging methods developed in 2D for the 3D task, this does seem to weaken this paper's contribution.
- The Hybrid Matching Strategy is also very similar to Semi-DETR's Stage-wise Hybrid Matching. While there is a difference - the proposed work explicitly does one-to-many matching during pre-training and one-to-one during semi-supervised, while Semi-DETR trains semi-supervised initially with one-to-many before switching to one-to-one, Semi-DETR does not have an explicit labeled-only pre-training phase, making the two pipelines functionally similar.
- The authors mention the necessity of comparing improvement, not absolute performance, due to the difference in base architectures. However, the proposed model has smaller improvement than baselines in Table 1, albeit it has a higher starting point, and it also does not achieve as good performance in SUN RGB-D.

**Questions:**

While this paper demonstrates strong performance in SSL 3D detection, it seems like most of the module contributions are brought from prior work Semi-DETR and 3DIoUMatch. While I believe that deriving best practices from previous methods is important and I also acknowledge the difficulties in transfering some 2D techniques to 3D, it also does seem to me that this paper's unique contributions may not be sufficient. At this stage, the differences between the proposed modules and prior work is not sufficiently discussed - if the authors can discuss this, I will consider raising my score. At this stage, I recommend a 2.

---

> ### Author Response · Authors · 2025-11-23
> **Response to Reviewer Yq3F**
>
> We thank the reviewer for the careful reading and the detailed, technically grounded comments. Below we address each weakness and question.
>
> **W1 / Q1 – RPLD vs 3DIoUMatch and other IoU-head SSL 3D methods**
>
> We agree that many recent SSL 3D detectors inherit the 3DIoUMatch idea of learning a 3D IoU predictor. Our point is not that “having an IoU head” is new, but that **how it is realized and used for 3D DETR is substantially different**.
>
> - **Where the features come from.** 3DIoUMatch, Diffusion-SS3D and DQS3D are built on proposal/voxel-based detectors. Their IoU head takes **pooled point/voxel features inside predicted boxes**, and the detector backbone itself is not query-based. In V-DETR, supervision is attached to a fixed set of **object queries** that already encode class, center, size and yaw. RPLD regresses IoU **directly from these decoder queries**, instead of re-aggregating points in each box. This is crucial because DETR queries do not carry an explicit point set, and forcing a 3DIoUMatch-style pooling layer on top of V-DETR breaks the native query abstraction.
>
> - **How the IoU prediction is used.** In 3DIoUMatch and its derivatives, the IoU head is tightly coupled with **NMS and test-time refinement**, and is used as an extra score in ranking. In Semi-3DETR, IoU is purely a **training-time, query-level signal**: it (i) gates teacher pseudo labels on unlabeled data, (ii) enters an IoU-aware Hungarian cost, and (iii) provides a geometric reliability term that QAC uses to decide which queries to align.
>
> To make this concrete, we plugged a **3DIoUMatch-style IoU head** into the same teacher–student framework and training schedule. Under the same V-DETR backbone, this variant significantly underperforms our query-driven RPLD design in Table 5. In other words, simply “reusing the 3DIoUMatch IoU module” is not enough in the 3D DETR setting, and the query-level design and its integration into matching/consistency matter for final mAP. We therefore see RPLD as a **non-trivial adaptation of the IoU-head idea to 3D DETR**, not a direct copy.
>
> **W2 – QAC vs Semi-DETR’s query consistency**
>
> Semi-DETR is formulated purely in **2D image space**. It mines pseudo 2D boxes, **crops backbone features** inside each box as RoI features, and applies a weak/strong consistency loss on these cropped 2D regions across views/augmentations.
>
> Semi-3DETR faces a different situation: V-DETR operates in **point clouds with metric 3D boxes**, and each query must represent a 3D object aggregated across views. Our Query Alignment Consistency (QAC) is therefore constructed in **3D query space**:
>
> - From each teacher pseudo box, we encode its **center, size and orientation** into the positional / reference embedding, and combine this with the decoder’s content query, to reconstruct a **3D object query** in V-DETR’s native representation.
>
> - The *same* reconstructed 3D queries are injected into both teacher and student decoders as anchors, and the consistency loss is applied directly on these 3D queries, not on 2D RoIs or image patches.
>
> Empirically, Table 4 shows that QAC yields consistent gains over the SSL baseline even when it is applied on all pseudo labels, including noisy ones. This indicates that the benefit is not coming from “generic consistency regularization” alone, but from **geometry-aware 3D query alignment that respects the V-DETR query parameterization**, which has no direct analogue in Semi-DETR.

---

> > ### Author Response · Authors · 2025-11-23
> >
> > **W3 – HMS vs Semi-DETR’s stage-wise hybrid matching**
> >
> > We share the motivation with Semi-DETR that mixing O2M and O2O is helpful for DETR optimization, but the **way it is scheduled and where it is applied** must change in 3D SSL.
> >
> > Semi-DETR adopts a **single-stage schedule**: for both labeled and unlabeled images, training starts with O2M and gradually anneals to O2O. This is well-behaved in 2D, where pseudo boxes are relatively accurate. In 3D, however, pseudo boxes can be severely wrong in depth and yaw, and O2M on unlabeled data tends to propagate a bad pseudo label to many queries.
> >
> > Our Hybrid Matching Strategy (HMS) therefore makes two deliberate choices:
> >
> > 1. Use **O2M only on clean supervised data** (pre-training) to stabilize query learning.
> > 2. Switch the student to **pure IoU-aware O2O in the SSL phase**, such that each pseudo 3D box supervises at most one query.
> >
> > The ScanNet 5% ablation clarifies this:
> >
> > | Variant                              | Pre-training matching | SSL matching        | Classifier | mAP@0.25 | mAP@0.5 |
> > |--------------------------------------|-----------------------|---------------------|-----------:|---------:|--------:|
> > | [A]                                  | O2O                   | O2O                 | softmax    |   52.7   |   35.9  |
> > | [B]                                  | O2M                   | O2M                 | softmax    |   23.6   |   14.2  |
> > | [C]                                  | O2M                   | O2O                 | sigmoid    |   13.4   |   11.4  |
> > | [D] **HMS (ours)**                   | O2M                   | O2O                 | softmax    | **54.1** | **39.5**|
> > | [E] Semi-DETR-style (O2M→O2O)        | O2M→O2O               | O2M→O2O             | sigmoid    |   13.9   |   10.7  |
> > | [F] Semi-DETR-style (O2M→O2O)        | O2M→O2O               | O2M→O2O             | softmax    |   47.4   |   32.8  |
> >
> > Key observations:
> >
> > - **[D] HMS** is the best: O2M is confined to clean labels, while SSL uses IoU-aware O2O plus softmax.
> > - **[B]** shows that running O2M on pseudo labels is highly destructive in 3D SSL.
> > - **[E]/[F]** mimic a Semi-DETR-style **single-stage O2M→O2O** schedule on both labeled and unlabeled data, and even with softmax ([F]) they remain clearly below HMS.
> > - **[C]** shares the O2M→O2O pattern with HMS but uses sigmoid. This collapses, highlighting that **the matching schedule and activation function need to be co-designed** for 3D IoU-aware matching.
> >
> > Therefore, while both works touch “hybrid matching”, HMS is a **different regime tailored to noisy 3D pseudo labels**: O2M is restricted to supervised pre-training, and the SSL phase is purely O2O with IoU-aware cost and a softmax classifier.
> >
> > **W4 – Improvement vs baselines and SUN RGB-D**
> >
> > We agree that comparing absolute mAP across heterogeneous backbones can be misleading, so we look at both **absolute performance** and **relative improvement over each model’s own backbone**.
> >
> > - On **ScanNet (Table 1)**, Semi-3DETR attains higher mAP than DQS3D at all label ratios. In the hardest **5% labeled** regime, the **gain over the supervised V-DETR baseline is larger** than the gain DQS3D brings over its FCAF3D baseline. This indicates that, once backbone differences are factored out, our SSL design is very effective at exploiting unlabeled data in the low-label regime.
> >
> > - On **SUN RGB-D (Table 2)**, DQS3D has higher mAP because it is built on FCAF3D, a very strong voxel-based backbone, while Semi-3DETR is built on V-DETR. For SUN RGB-D, since the original V-DETR code is only released on ScanNet, we had to re-implement V-DETR following 3DETR, which yields a weaker supervised baseline and partly explains the gap. However, again at **5% labels**, Semi-3DETR gives a **larger relative gain over this V-DETR baseline** than DQS3D gives over FCAF3D.
> >
> > We see Semi-3DETR as the **first DETR-style SSL 3D detector**. Its goal is not to supplant all voxel/vote methods, but to show that a query-based 3D DETR can make competitive use of unlabeled indoor scenes. The consistent gains over a strong V-DETR backbone, especially at 5% labels, suggest that this direction is technically meaningful rather than a minor re-combination of existing 2D/3D ideas.

---

### Author Response · Authors · 2025-11-23
**General Response**

We thank the reviewers for their careful reading and constructive comments. The main concerns center on:

- **(i)** how the proposed **RPLD**, **QAC**, and **HMS** differ from **3DIoUMatch** and **Semi-DETR**;
- **(ii)** justification and empirical support for key design choices, e.g. query-driven IoU head and query reconstruction, hybrid matching schedule, softmax vs. sigmoid;
- **(iii)** sensitivity to thresholds and consistency weights, e.g. τ, λ;
- **(iv)** the focus on indoor **ScanNet/SUN RGB-D** without **KITTI/nuScenes**; and
- **(v)** how to interpret our gains relative to backbone differences and to **DQS3D** on SUN RGB-D.

In the detailed responses, we clarify that our contributions are not simply adding an IoU head or reusing 2D SSL machinery, but **adapting these ideas to query-based 3D DETR** with dedicated designs and tight integration into matching and consistency. We connect these design choices directly to the reported ablations (IoU-head variants, matching schedules, activations, and QAC), explain the computational overhead relative to V-DETR, and summarize the robustness to **τ** and **λ**, using a **single configuration across datasets and label ratios**. We also make explicit our evaluation protocol (labeled subsets and use of unlabeled data) and discuss why we report both absolute mAP and **relative gains over each backbone**, as well as the **complementary role** of Semi-3DETR to voxel-based methods such as DQS3D.

These clarifications will be incorporated into the revised manuscript to make the **novelty, scope, and empirical evidence** of Semi-3DETR more explicit.

---

### Note · Authors · 2026-02-09

I have read and agree with the venue's withdrawal policy on behalf of myself and my co-authors.

---

### Meta-Review · Area_Chair_d7g9 · 2025-12-24

**Summary:**

This paper studies semi-supervised 3D object detection and proposes Semi-3DETR, a framework that extends DETR-style 3D detectors to the semi-supervised setting via three components: Robust Pseudo-Label Denoising (RPLD), Query Alignment Consistency (QAC), and a Hybrid Matching Strategy (HMS). Reviewers generally agree that the paper is well written, the problem is timely, and the empirical results are encouraging. Nevertheless, after considering the rebuttal and post-discussion feedback, substantial concerns remain regarding the degree of methodological novelty and the extent to which the proposed components are fundamentally distinct from prior semi-supervised 3D detection and 2D Semi-DETR techniques. In particular, several reviewers remain unconvinced that RPLD, QAC, and HMS go beyond careful adaptations of existing IoU-head, consistency, and hybrid matching ideas, rather than constituting clearly new algorithmic contributions. Additional concerns include limited dataset coverage (indoor-only benchmarks), sensitivity to design choices and thresholds, and mixed absolute performance compared to strong voxel-based baselines on SUN RGB-D. While the work is technically sound and thoughtfully engineered, these unresolved issues prevent it from meeting the bar for a clear acceptance at a top-tier venue in its current form.

**Reviewer Concerns:**

Reviewer Yq3F: Remains unconvinced that RPLD, QAC, and HMS provide sufficient methodological novelty beyond prior work such as 3DIoUMatch and Semi-DETR, despite the rebuttal clarifications.

Reviewer Yssm: Appreciates the motivation and design but raises concerns about limited dataset coverage, sensitivity to design choices, and mixed absolute performance compared to strong voxel-based baselines.

Reviewer BZnT: Finds the experiments and ablations convincing, but questions the generalization of the method beyond indoor scenes and the necessity of the proposed framework given existing alternatives.

Reviewer Sh9H: Generally positive about the empirical results and presentation, but remains cautious about the overall contribution level and incremental nature of the proposed approach.

**Reviewer Scores:**

Reviewer Yq3F: Likely no change (reject).

Reviewer Yssm: Possibly a slight increase, but still below the acceptance threshold.

Reviewer BZnT: Likely no change (borderline accept / weak accept).

Reviewer Sh9H: Likely no change or a slight increase.

---

### Decision · Program_Chairs · 2026-01-26

Reject